# A Newcastle Disease Virus (NDV) Expressing a Membrane-Anchored Spike as a Cost-Effective Inactivated SARS-CoV-2 Vaccine

**DOI:** 10.3390/vaccines8040771

**Published:** 2020-12-17

**Authors:** Weina Sun, Stephen McCroskery, Wen-Chun Liu, Sarah R. Leist, Yonghong Liu, Randy A. Albrecht, Stefan Slamanig, Justine Oliva, Fatima Amanat, Alexandra Schäfer, Kenneth H. Dinnon, Bruce L. Innis, Adolfo García-Sastre, Florian Krammer, Ralph S. Baric, Peter Palese

**Affiliations:** 1Department of Microbiology, Icahn School of Medicine at Mount Sinai, New York, NY 10029, USA; weina.sun@mssm.edu (W.S.); stephen.mccroskery@icahn.mssm.edu (S.M.); wenchun0617@gmail.com (W.-C.L.); yonghong.liu@mssm.edu (Y.L.); randy.albrecht@mssm.edu (R.A.A.); st.slamanig@gmail.com (S.S.); justine.oliva@mssm.edu (J.O.); fatima.amanat@icahn.mssm.edu (F.A.); Adolfo.Garcia-Sastre@mssm.edu (A.G.-S.); florian.krammer@mssm.edu (F.K.); 2Global Health Emerging Pathogens Institute, Icahn School of Medicine at Mount Sinai, New York, NY 10029, USA; 3Biomedical Translation Research Center, Academia Sinica, Taipei 11571, Taiwan; 4Department of Epidemiology, University of North Carolina at Chapel Hill, Chapel Hill, NC 27599, USA; leist@email.unc.edu (S.R.L.); aschaefe@email.unc.edu (A.S.); rbaric@email.unc.edu (R.S.B.); 5Graduate School of Biomedical Sciences, Icahn School of Medicine at Mount Sinai, New York, NY 10029, USA; 6Department of Microbiology and Immunology, University of North Carolina at Chapel Hill, Chapel Hill, NC 27599, USA; kdinnon@email.unc.edu; 7PATH, Washington, DC 20001, USA; binnis@path.org; 8Department of Medicine, Icahn School of Medicine at Mount Sinai, New York, NY 10029, USA; 9The Tisch Cancer Institute, Icahn School of Medicine at Mount Sinai, New York, NY 10029, USA

**Keywords:** egg-based vaccine, adjuvant, antigen-sparing, mouse-adapted SARS-CoV-2, hamster model, COVID-19

## Abstract

A successful severe acute respiratory syndrome coronavirus 2 (SARS-CoV-2) vaccine must not only be safe and protective, but must also meet the demand on a global scale at a low cost. Using the current influenza virus vaccine production capacity to manufacture an egg-based inactivated Newcastle disease virus (NDV)/SARS-CoV-2 vaccine would meet that challenge. Here, we report pre-clinical evaluations of an inactivated NDV chimera stably expressing the membrane-anchored form of the spike (NDV-S) as a potent coronavirus disease 2019 (COVID-19) vaccine in mice and hamsters. The inactivated NDV-S vaccine was immunogenic, inducing strong binding and/or neutralizing antibodies in both animal models. More importantly, the inactivated NDV-S vaccine protected animals from SARS-CoV-2 infections. In the presence of an adjuvant, antigen-sparing could be achieved, which would further reduce the cost while maintaining the protective efficacy of the vaccine.

## 1. Introduction

A severe acute respiratory syndrome coronavirus 2 (SARS-CoV-2) vaccine is urgently needed to mitigate the current coronavirus disease 2019 (COVID-19) pandemic worldwide. Numerous vaccine approaches are being developed [1,2,3,4]. However, many of them are not likely to be cost-effective and affordable for low-income countries and under-insured populations. This could be of concern in the long run, as it is crucial to vaccinate a larger population than the high-income minority to establish herd immunity and effectively contain the spread of the virus. Among all the SARS-CoV-2 vaccine candidates, an inactivated vaccine might have the advantage over live vaccines of having a better safety profile in vulnerable individuals. In addition, inactivated vaccines could be combined with an adjuvant to obtain a better protective efficacy and dose-sparing to meet the large global demand. However, the current platform for producing the inactivated whole virion SARS-CoV-2 vaccine requires propagation of the virus in cell culture under biosafety level 3 (BSL-3) conditions and only very few BSL-3 vaccine production facilities exist [3], limiting the scaling. Excessive inactivation procedures might have to be implemented to ensure complete inactivation of the virus, at the risk of losing antigenicity of the vaccine. Many viral vector vaccines against coronaviruses have been developed, most of which are used as live vaccines [4,5,6,7,8,9]. In addition, the efficacy of certain viral vectors could be dampened by pre-existing immunity to the viral backbone in the human population. Most recombinant protein vaccines require cumbersome manufacturing procedures that would make it difficult to conduct inexpensive mass manufacturing. Genetic vaccines (mRNA and DNA vaccines) display great promise, but as they have only recently been developed, their performance in humans is uncertain.

We have previously reported the construction of Newcastle disease virus (NDV)-based viral vectors expressing an uncleaved spike protein, whose transmembrane domain and cytoplasmic tail were replaced with those from the NDV fusion (F) protein (S-F chimera) [10]. We have shown that these NDV vector vaccines grow well in embryonated chicken eggs, and that the SARS-CoV-2 spike (S) proteins are abundantly incorporated into the NDV virions [10]. The NDV vector, based on a vaccine virus strain against an avian pathogen, overcomes the abovementioned limitation for viral vector vaccines and allows the manufacturing of the vaccine under BSL-2 conditions, prior to its inactivation. The construct used as the inactivated vaccine in this study expresses an S-F chimera and has a mutation (L289A) in the F protein of NDV (NDV-S), which was shown to facilitate hemagglutinin-neuraminidase (HN)-independent fusion of the virus (Figure 1A). This mutant NDV (F L289A) is currently being used as an oncolytic agent in a Phase I trial (NCT04135352). To develop an NDV-based inactivated SARS-CoV-2 vaccine, the existing global influenza virus vaccine production capacity could be employed, with minor modifications to the manufacturing pipeline of inactivated influenza virus vaccines. As egg-grown influenza virus vaccines are inactivated by formalin or beta-propiolactone (BPL) treatment, we chose BPL inactivation for the NDV-S vaccine because it is believed to be a less disrupting process. Such inactivated NDV-S vaccines will display SARS-CoV-2 spike proteins, together with HN and F NDV proteins, on the surface of the whole inactivated virions. The inactivated NDV-S vaccine could be administered intramuscularly, with an adjuvant for dose sparing. This approach should be suitable for safely inducing spike-specific protective antibodies (Figure 1B).

In this study, we investigated NDV-S as an inactivated SARS-CoV-2 vaccine candidate with and without an adjuvant in mice and hamsters. We found that the S-F chimera expressed by the NDV vector is very stable, with no measurable loss of stability after 3 weeks of 4 °C storage in allantoic fluid. The beta-propiolactone (BPL)-inactivated NDV-S vaccine is immunogenic, inducing high titers of S-specific antibodies in both animal models. Furthermore, the effects of a clinical-stage investigational liposomal suspension adjuvant (R-enantiomer of the cationic lipid DOTAP (R-DOTAP)) [11,12,13,14], as well as an MF-59-like oil-in-water emulsion adjuvant (AddaVax), were also evaluated in mice. Both adjuvants were shown to achieve dose sparing (>10 fold) in mice. The vaccinated animals displayed less weight loss and significantly reduced viral loads in the lungs compared to the vector-only control animals after being challenged with a mouse-adapted SARS-CoV-2 strain. This is encouraging as the existing global egg-based production capacity for inactivated influenza virus vaccines could be utilized immediately to rapidly produce an egg-based NDV-S vaccine with minimal modifications to their production pipelines. Most importantly, this class of products is amenable to large-scale production at a low cost [15,16,17]. Alternatively, the NDV-S and other chimeric NDV vaccines could also be manufactured in cultured cells such as Vero cells [18].

## 2. Materials and Methods

### 2.1. Ethics Statement

Animal studies were performed in accordance with protocols (PROTO202000098 and CEIRS program, 13-0386 PRYR II-IACUC-2013-1408) approved by the Institutional Animal Care and Use Committee (IACUC) at the Icahn School of Medicine at Mount Sinai. All animals were housed in a temperature-controlled biosafety level 2 (BSL-2) animal facility in the Annenberg building and Icahn building. All efforts were made to minimize animal suffering.

### 2.2. Plasmids

The construction of the NDV_LS/L289A_S-F rescue plasmid has been described in a previous study [10]. Briefly, the sequence of the ectodomain of the S without the polybasic cleavage site (^682^RRAR^685^ to A) was amplified from the pCAGGS plasmid encoding the codon-optimized nucleotide sequence of the S gene (GenBank: MN908947.3) of a SARS-CoV-2 isolate (Wuhan-Hu-1/2020) by a polymerase chain reaction (PCR) [19], using primers containing the gene end (GE), gene start (GS), and Kozak sequences at the 5’ end [20]. The nucleotide sequence of the transmembrane domain (TM) and the cytoplasmic tail (CT) of the NDV_LaSota fusion (F) protein was codon-optimized for mammalian cells and synthesized by Integrated DNA Technologies (IDT, Coralville, IA, USA) (gBlock). The amplified S ectodomain was fused to the TM/CT of F through a GS linker (GGGGS). Additional nucleotides were added at the 3’ end to follow the “rule of six” of the paramyxovirus genome. The S-F gene was inserted between the P and M gene of the pNDV_LaSota (LS) L289A mutant (NDV_LS/L289A) antigenomic cDNA by in-Fusion cloning (Takara Bio USA Inc., Mountain View, CA, USA). This NDV_LS/L289A mutant is currently being used in a Phase I oncolytic NDV trial (NCT0413532). The recombination product was transformed into NEB^®^ Stable Competent *Escherichia. coli* (New England Biolabs Inc., Ipswich, MA, USA) to generate the NDV_LS/L289A_S-F rescue plasmid. The plasmid was purified using the PureLink^TM^ HiPure Plasmid Maxiprep Kit (Thermo Fisher Scientific, Waltham, MA, USA).

### 2.3. Cells and Viruses

BSRT7 cells stably expressing the T7 polymerase were kindly provided by Dr. Benhur Lee at ISMMS. The cells were maintained in Dulbecco’s Modified Eagle’s Medium (DMEM; Gibco, Gaithersburg, MA, USA) containing 10% (vol/vol) fetal bovine serum (FBS), 100 unit/mL of penicillin, and 100 µg/mL of streptomycin (P/S; Gibco) at 37 °C with 5% CO_2_. SARS-CoV-2 isolate USA-WA1/2020 (WA-1, BEI Resources NR-52281) used for hamster challenge was propagated in Vero E6 cells (ATCC CRL-1586) in Dulbecco’s Modified Eagle Medium (DMEM), supplemented with 2% fetal bovine serum (FBS), 4.5 g/L D-glucose, 4 mM L-glutamine, 10 mM Non-Essential Amino Acids, 1 mM Sodium Pyruvate, and 10 mM HEPES at 37 °C. All experiments with live SARS-CoV-2 were performed in the Centers for Disease Control and Prevention (CDC)/US Department of Agriculture (USDA)-approved biosafety level 3 (BSL-3) biocontainment facility of the Global Health and Emerging Pathogens Institute at the Icahn School of Medicine at Mount Sinai, in accordance with institutional biosafety requirements.

### 2.4. Rescue of NDV LaSota Expressing the Spike of SARS-CoV-2

To rescue NDV_LS/L289A_S-F, six-well plates of BSRT7 cells were seeded 3 × 10^5^ cells per well the day before transfection. The next day, 4 µg of pNDV_LS/L289A_S-F, 2 µg of pTM1-NP, 1 µg of pTM1-P, 1 µg of pTM1-L, and 2 µg of pCI-T7opt were re-suspended in 250 µL of Opti-MEM (Gibco, Gaithersburg, MA, USA). The plasmid cocktail was then gently mixed with 30 µL of TransIT LT1 transfection reagent (Mirus) [10]. The mixture was incubated at room temperature (RT) for 30 min. Toward the end of the incubation, the growth medium of each well was replaced with 1 mL of Opti-MEM. The transfection complex was added dropwise to each well and the plates were incubated at 37 °C with 5% CO_2_. The supernatant and cells from transfected wells were harvested at 48 h post-transfection, and briefly homogenized by several strokes using an insulin syringe. Two hundred microliters of the homogenized mixture was injected into the allantoic cavity of 8- to 10-day-old specific-pathogen-free (SPF) embryonated chicken eggs. The eggs were incubated at 37 °C for 3 days, before being cooled at 4 °C overnight. The allantoic fluid was collected and clarified by centrifugation. The rescue of NDV was determined by a hemagglutination (HA) assay using 0.5% chicken or turkey red blood cells. The RNA of the positive samples was extracted and treated with DNase I (Thermo Fisher Scientific, Waltham, MA, USA). A reverse transcriptase-polymerase chain reaction (RT-PCR) was performed to amplify the transgene. The sequences of the transgenes were confirmed by Sanger Sequencing (Genewiz, South Plainfield, NJ USA). Recombinant DNA experiments were performed in accordance with protocols approved by the Icahn School of Medicine at Mount Sinai Institutional Biosafety Committee (IBC).

### 2.5. Preparation of Concentrated Virus

Before concentrating the virus, allantoic fluids were clarified by centrifugation at 3441× *g* using a Sorvall Legend RT Plus Refrigerated Benchtop Centrifuge (Thermo Fisher Scientific, Waltham, MA, USA) at 4 °C for 30 min to remove debris. Live virus in the allantoic fluid was pelleted through a 20% sucrose cushion in NTE buffer (100 mM NaCl, 10 mM Tris-HCl, 1 mM EDTA, pH 7.4) by ultra-centrifugation in a Beckman L7-65 ultracentrifuge at 25,000 rpm for two hours at 4 °C using a Beckman SW28 rotor (Beckman Coulter, Brea, CA, USA). Supernatants were aspirated off and the pellets were re-suspended in PBS (pH 7.4). The protein content was determined using the bicinchoninic acid (BCA) assay (Thermo Fisher Scientific, Waltham, MA, USA). To prepare inactivated concentrated viruses, one part of 0.5 M disodium phosphate (DSP) was mixed with 38 parts of the allantoic fluid to stabilize the pH. One part of 2% beta-propiolactone (BPL) was added dropwise to the mixture during shaking, which gave a final concentration of 0.05% BPL. The treated allantoic fluid was mixed thoroughly and incubated on ice for 30 min. The mixture was then placed in a 37 °C water bath shaken every 15 min for two hours. The inactivated allantoic fluid was clarified by centrifugation at 3441× *g* for 30 min. The loss of infectivity was confirmed by the lack of growth (determined by the HA assay) of the virus (1:1000 dilution in PBS) in 10-day-old embryonated chicken eggs that were inoculated. The inactivated viruses were concentrated as described above.

### 2.6. Evaluation of the Stability of the S-F in the Allantoic Fluid

The allantoic fluid containing the NDV_LS/L289A_S-F virus was harvested and clarified by centrifugation. The clarified allantoic fluid was aliquoted into 15 mL volumes. Week (wk) 0 allantoic fluid was concentrated immediately after centrifugation as described above, through a 20% sucrose cushion. The pelleted virus was re-suspended in 300 µL phosphate buffered saline (PBS) and stored at −80 °C. The other three aliquots of the allantoic fluid were maintained at 4 °C to test the stability of the S-F construct. Week 1, 2, and 3 samples were collected consecutively on a weekly basis, and concentrated virus was prepared in 300 µL PBS using the same method. The protein content of the concentrated virus from wk 0, 1, 2, and 3 was determined using the BCA assay after one freeze-thaw from −80 °C. One microgram of each concentrated virus preparation was resolved in 4–20% sodium dodecyl sulfate polyacrylamide gel electrophoresis (SDS-PAGE; Bio-Rad, Hercules, CA, USA). The S-F protein and the NDV hemagglutinin-neuraminidase (HN) protein were detected by Western blot.

### 2.7. Western Blot

Concentrated live or inactivated virus samples were mixed with Novex™ Tris-Glycine SDS Sample Buffer (2X) (Thermo Fisher Scientific, Waltham, MA, USA) with NuPAGE™ Sample Reducing Agent (10X) (Thermo Fisher Scientific, Waltham, MA, USA). One or two micrograms of the concentrated viruses was heated at 95 °C for 5 min, before being resolved on 4–20% SDS-PAGE (Bio-Rad, Hercules, CA, USA), using the Novex™ Sharp Pre-stained Protein Standard (Thermo Fisher Scientific, Waltham, MA, USA) as the protein marker. To perform Western blots, proteins were transferred onto a polyvinylidene difluoride (PVDF) membrane (GE healthcare, Chicago, IL USA). The membrane was blocked with 5% non-fat dry milk in PBS containing 0.1% *v/v* Tween 20 (PBST) for 1 h at room temperature (RT). The membrane was washed with PBST on a shaker three times (10 min at RT each time) and incubated with an S-specific mouse monoclonal antibody 2B3E5 (provided by Dr. Thomas Moran at ISMMS) or an HN-specific mouse monoclonal antibody 8H2 (MCA2822, Bio-Rad, Hercules, CA, USA) diluted in PBST containing 1% bovine serum albumin (BSA) overnight at 4 °C. The membranes were then washed with PBST on a shaker three times (10 min at RT each time) and incubated with secondary sheep anti-mouse IgG linked with horseradish peroxidase (HRP) diluted (1:2000) in PBST containing 5% non-fat dry milk. The secondary antibody was discarded and the membranes were washed with PBST on a shaker three times (10 min at RT each time). Pierce™ ECL Western Blotting Substrate (Thermo Fisher Scientific, Waltham, MA, USA) was added to the membrane, and the blots were imaged using the Bio-Rad Universal Hood Ii Molecular imager (Bio-Rad, Hercules, CA, USA) and processed by Image Lab Software (Bio-Rad, Hercules, CA, USA).

### 2.8. Immunization and Challenge Study in BALB/c Mice

Seven-week-old female BALB/cJ mice (Jackson Laboratories, Bar Harbor, ME, USA) were used in this study. Experiments were performed in accordance with protocols approved by the Icahn School of Medicine at Mount Sinai Institutional Animal Care and Use Committee (IACUC). Mice were divided into 10 groups (*n* = 5) receiving the inactivated virus without or with an adjuvant at three different doses intramuscularly. The vaccination followed a prime-boost regimen with a two-week interval. Specifically, group 1, group 2, and group 3 received 5, 10, and 20 µg inactivated NDV-S vaccine (total protein) without the adjuvant, respectively; group 4, group 5, and group 6 received low doses of 0.2, 1, and 5 µg inactivated NDV-S vaccine, respectively, combined with 300 µg/mouse of R-DOTAP (PDS Biotechnology); group 7, group 8, and group 9 mice received 0.2, 1, and 5 µg inactivated NDV-S vaccine, respectively, with 50 µL/mouse of AddaVax (Invivogen) as the adjuvant; and group 10 received 20 µg inactivated wild type (WT) NDV as the vector-only control. The SARS-CoV-2 challenge was performed at the University of North Carolina by Dr. Ralph Baric’s group in a biosafety level 3 (BSL-3) facility. Mice were intranasally (i.n.) challenged 19 days after the boost using a mouse-adapted SARS-CoV-2 strain at 7.5 × 10^4^ plaque forming units (PFU) [2,21]. Weight loss was monitored for 4 days.

### 2.9. Immunization and Challenge Study in Golden Syrian Hamsters

Eight-week-old female golden Syrian hamsters were used in this study. Experiments were performed in accordance with protocols approved by the Icahn School of Medicine at Mount Sinai Institutional Animal Care and Use Committee (IACUC). Four groups (*n* = 8) of hamsters were included. The inactivated vaccines were given intramuscularly following a prime-boost regimen with a two-week interval. Group 1 received 10 µg of inactivated NDV-S vaccine, group 2 received 5 µg of inactivated NDV-S vaccine combined with 50 µL of AddaVax per hamster, and group 3 hamsters received 10 µg of inactivated WT NDV as the vector-only control. A healthy control group receiving no vaccine was also included. Twenty-four days after the boost, hamsters were challenged intranasally with 10^4^ PFU of the USA-WA1/2020 SARS-CoV-2 strain in a biosafety level 3 (BSL-3) facility. Weight loss was monitored for 5 days.

### 2.10. Lung Titers

The inferior lung lobes of mice were collected and homogenized in 1 mL PBS. Upper right (UR) and lower right (LR) lung lobes of hamsters were harvested at day 2 and day 5 post-infection. Each lung lobe of hamsters was homogenized in 1 mL PBS. A plaque assay was performed to measure the viral titer in the lung homogenates, as described previously [2,10,21]. Geometric mean titers of plaque forming units (PFU) per lobe (mice) or per mL (hamsters) were calculated using GraphPad Prism 7.0.

### 2.11. Enzyme Linked Immunosorbent Assays (ELISAs)

Mice were bled pre-boost and 11 days after the boost. Hamsters were bled pre-boost and 26 days after the boost. Sera were isolated by low-speed centrifugation. ELISAs were performed as described previously [19]. Briefly, Immulon 4 HBX 96-well ELISA plates (Thermo Fisher Scientific, Waltham, MA, USA) were coated with 2 µg/mL of recombinant trimeric S protein produced in insect cells (50 µL per well) in coating buffer (SeraCare Life Sciences Inc., Milford, MA, USA) overnight at 4 °C [19]. The next day, all plates were washed three times with 220 µL PBS containing 0.1% (*v*/*v*) Tween-20 (PBST) and blocked in 220 µL blocking solution (3% goat serum, 0.5% non-fat dried milk powder, 96.5% PBST) for 1 h at RT. Both mouse sera and hamster sera were three-fold serially diluted in blocking solution starting at 1:30, followed by a 2 h incubation at RT. ELISA plates were washed three times with PBST and incubated in 50 µL per well of sheep anti-mouse IgG-horseradish peroxidase (HRP) conjugated antibody (GE Healthcare, Chicago, IL USA) or goat anti-hamster IgG-HRP conjugated antibody (Invitrogen, Carlsbad, CA, USA) diluted (1:3000) in blocking solution. Plates were washed three times with PBST and 100 µL of *o*-phenylenediamine dihydrochloride (SigmaFast OPD; Sigma, St. Louis, MO, USA) substrate was added per well. After developing the plates for 10 min, 50 µL of 3 M hydrochloric acid (HCl) was added to each well to stop the reactions. The optical density (OD) was measured at 492 nm on a Synergy 4 plate reader (BioTek, Winooski, VT, USA) or equivalents. An average of OD values for blank wells plus three standard deviations was used to set a cutoff for plate blank outliers. A cutoff value was established for each plate that was used for calculating the endpoint titers. The endpoint titers of serum IgG responses were graphed using GraphPad Prism 7.0.

### 2.12. Micro-Neutralization Assay

All neutralization assays were performed using Vero E6 cells in the biosafety level 3 (BSL-3) facility following institutional guidelines, as described previously [19,22]. Serum samples were heat-inactivated at 56 °C for 60 min prior to use. Pooled sera in technical duplicates were serially diluted three-fold, starting at 1:20 dilution. The cells were fixed with 100 µL 10% formaldehyde per well for 24 h, before being taken out of the BSL-3 facility. A cell-based ELISA using an anti-NP antibody (1C7), kindly provided by Dr. Thomas Moran at ISMMS, was performed in a BSL-2 biosafety cabinet, as previously described [19,22]. The OD of 492 nm was measured on a Biotek SynergyH1 Microplate Reader. Non-linear regression curve fit analysis (the top and bottom constraints were set at 100% and 0%, respectively) over the dilution curve was performed to calculate 50% of inhibitory dilution (ID_50_) of the serum using GraphPad Prism 7.0.

### 2.13. Statistics

The statistical analysis was performed using GraphPad Prism 7.0. The statistical difference in lung viral titers was determined using the Kruskal–Wallis test with Dunn’s correction for multiple comparisons.

## 3. Results

### 3.1. The Spike Protein Expressed on NDV Virions Is Stable in Allantoic Fluid

The stability of the antigen could be of concern as the vaccine needs to be purified and inactivated through a temperature-controlled (~4 °C) process. The final product is often formulated and stored in liquid buffer at 4 °C. To examine the stability of the S-F protein, allantoic fluid containing the NDV-S live virus was aliquoted into equal volumes (15 mL) and stored at 4 °C. Samples were collected weekly (wk 0, 1, 2, and 3) and concentrated through a 20% sucrose cushion. The concentrated virus was re-suspended in equal amounts of PBS. The total protein content of the four aliquots was comparable among the preparations (wk 0: 0.94 mg/mL; wk 1: 1.04 mg/mL; wk 2: 0.9 mg/mL; wk 3: 1.08 mg/mL). The stability of the S-F construct was evaluated by Western blot with the anti-S monoclonal antibody 2B3E5. Compared to the stability of the NDV HN protein, the Spike protein remained stable when kept in allantoic fluid at 4 °C (Figure 2A). The inactivation by 0.05% BPL was confirmed by the lack of HA activity following inoculation of the inactivated virus into embryonated chicken eggs (Figure 2B). Importantly, the inactivation procedure using 0.05% BPL did not cause any loss of antigenicity of the S-F, as evaluated by Western blot (Figure 2C). These observations demonstrate that the membrane-anchored S-F chimera expressed by the NDV vector is very stable, without degradation at 4 °C for 3 weeks or when treated with BPL for inactivation.

### 3.2. Inactivated NDV-S Vaccine Induced High Titers of Binding and Neutralizing Antibodies in Mice

For a pre-clinical evaluation of the inactivated NDV-S vaccine, the immunogenicity and dose-sparing ability of the adjuvants were investigated in mice. A dose-ranging study of the vaccine in the presence or absence of an adjuvant was evaluated based on the ability to induce antibody/neutralizing antibody responses. After partial purification, the vaccine preparation was administered intramuscularly, following a prime-boost regimen with a 2-week interval. Specifically, for the three unadjuvanted groups, mice were intramuscularly immunized with increasing doses of inactivated NDV-S vaccine at 5, 10, or 20 μg per mouse. Two adjuvants were tested here: A clinical-stage adjuvant, liposomal suspension of the pure R-enantiomer of the cationic lipid DOTAP (R-DOTAP) and the MF59-like oil-in-water emulsion adjuvant AddaVax. Each adjuvant was combined with low doses of NDV-S vaccines at 0.2, 1, and 5 μg. Mice receiving 20 μg of inactivated WT NDV were used as vector-only (negative) controls (Figure 3A). Mice were bled pre-boost (2 weeks after prime) and 11 days post-boost to examine antibody responses by ELISAs and micro-neutralization assays (Figure 3A) [19,22]. After one immunization, all vaccination groups developed S-specific antibodies. The boost greatly increased the antibody titers of all NDV-S immunization groups. Immunization with R-DOTAP combined with 5 μg of vaccine induced the highest antibody titer. Immunization with one microgram of vaccine formulated with R-DOTAP or AddaVax and 5 μg of vaccine with AddaVax induced comparable levels of binding antibody, which is also similar to the titers induced by 20 μg of vaccine without an adjuvant. As expected, immunization with the inactivated wild-type NDV virus did not induce S-specific antibody responses (Figure 3B). We performed microneutralization assays to determine the neutralizing activity of serum antibodies collected from vaccinated mice. Except for mice immunized with the WT NDV, sera from all mice immunized with the NDV-S vaccine showed neutralizing activity against the SARS-CoV-2 USA-WA1/2020 strain. The neutralization titers induced by the immunization of 1 μg of vaccine with R-DOTAP (ID_50_ of ~476) and 5 μg of vaccine with AddaVax groups (ID_50_ of ~515) appeared to be the highest and were comparable to each other. These levels are also in the higher range of human convalescent serum neutralization titers, as measured in our previous studies [19,23]. Interestingly, although the group receiving 5 μg of vaccine with R-DOTAP developed the most abundant binding antibodies detected by ELISA, these sera were not the most neutralizing ones, suggesting that R-DOTAP might have a different impact on immunogenicity compared to AddaVax. It is possible that with more antigen combined with R-DOTAP, the immune responses were skewed towards the induction of non-neutralizing antibodies (Figure 3C). In any case, these results demonstrated that the inactivated NDV-S vaccine expressing the membrane-anchored S-F was immunogenic, inducing potent binding and neutralizing antibodies. Importantly, at least 10-fold dose sparing was achieved with an adjuvant in mice.

### 3.3. The Inactivated NDV-S Vaccine Protects Mice from Infection by a Mouse-Adapted SARS-CoV-2 Virus

To evaluate vaccine-induced protection, mice were challenged 19 days post-boost using a mouse-adapted SARS-CoV-2 virus (Figure 3A) [2,10,21]. Weight loss was monitored for 4 days post-infection, at which point the mice were euthanized to assess pulmonary virus titers. Only the negative control group receiving the WT NDV, was observed to lose notable weight (~10%) by day 4 post-infection, while all of the vaccinated groups showed no weight loss (Figure 4A). Viral titers in the lung at 4 days post-challenge were also measured. As expected, the negative control group given the WT NDV exhibited the highest viral titer of >10^4^ PFU/lobe. Groups receiving 5 μg of unadjuvanted vaccine or 0.2 μg of vaccine with R-DOTAP exhibited detectable but low viral titers in the lung, while all of the other groups were fully protected, showing no viral loads (Figure 4B). These results are encouraging as immunization with 0.2 μg of vaccine adjuvanted with AddaVax conferred a level of protection that was equal to that induced by immunization with 10 μg of vaccine without an adjuvant. Although 0.2 μg of vaccine with R-DOTAP did not induce sterilizing immunity, approximately, a 1000-fold reduction of viral titer in the lungs was achieved.

### 3.4. The Inactivated NDV-S Vaccine Confers Protection against SARS-CoV-2 Infection in a Hamster Model

Golden Syrian hamsters have been characterized as a useful small animal model for COVID-19 as they are susceptible to SARS-CoV-2 infections and manifest SARS-CoV-2-induced diseases [24,25]. Here, we conducted a pilot study that assessed the immunogenicity and protective efficacy of the inactivated NDV-S vaccine in hamsters. Female golden Syrian hamsters were immunized by a prime-boost regimen with a 2-week interval via the intramuscular administration route. Twenty-four days after the booster immunization, hamsters were intranasally infected with 10^4^ PFU of SARS-CoV-2 (USA-WA1/2020) virus. Animals were bled pre-boost and at 2 days post-infection (dpi). Lungs of a subset of animals were harvested at 2 dpi. The lungs of the rest of the animals were harvested at 5 dpi. Four groups of hamsters were included in this pilot study. Group 1 was immunized with 10 μg of inactivated NDV-S vaccine per animal without adjuvants. Group 2 received 5 μg of inactivated NDV-S vaccine with AddaVax as an adjuvant. Group 3 was immunized with 10 μg of inactivated WT NDV as the vector-only negative control group. Group 4, which was not vaccinated and was mock-challenged with PBS, was included as the healthy control group (Figure 5A). Serum IgG titers sampled prior to the booster immunization and at 2 dpi were measured by ELISA. One immunization with NDV-S vaccine with or without the adjuvant successfully induced spike-specific antibodies. Since there was no seroconversion from infection at 2 dpi indicated by the baseline level of the WT NDV sera, the increase in titers at 2 dpi compared to titers after vaccine priming most likely represents vaccine-induced antibody levels after the boost. As expected, the boost substantially increased the antibody titers in the NDV-S vaccination groups, whereas the WT NDV sera showed negligible binding signals (Figure 5B). Nevertheless, we cannot exclude a contribution from a rapid production of S antibodies by vaccine-induced memory B cells after exposure to SARS-CoV-2. Hamsters were challenged and weight loss was monitored for 5 days. The WT NDV group lost up to 15% of its weight by 5 dpi. Animals receiving 10 μg of inactivated NDV-S vaccine lost ~10% of their weight by 3 dpi, at which point body weights started to recover. Animals receiving 5 μg of inactivated NDV-S vaccine with AddaVax only lost weight by 2 dpi, at which point body weights started to recover (Figure 5C). Viral titers in the upper right (UR) lung lobes and lower right (LR) lung lobes were also measured. The lung lobes were homogenized in 1 mL of PBS. Viral titers in the lung homogenates were measured by a plaque assay. Animals vaccinated with NDV-S with or without adjuvant displayed a substantial reduction of viral titers at 2 dpi, while the viral titers of these two groups at 5 dpi were below the limit of detection (Figure 5D).

## 4. Discussion

We have previously reported NDV-based SARS-CoV-2 live vaccines expressing two forms of spike protein (S and S-F) [10]. Since the S-F showed superior incorporation into NDV particles, we investigated its potential to be used as an inactivated vaccine in this study. The NDV-S was found to be very stable when stored at 4 °C for 3 weeks, without degradation of the S-F protein. In mice, we have shown that a total amount of inactivated NDV-S vaccine as low as 0.2 μg could significantly reduce viral titers in the lung when combined with R-DOTAP, by approximately a factor of 1000, while the adjuvant AddaVax conferred even better protection. The NDV-S vaccine at 1 μg with either adjuvant elicited potent neutralizing antibodies and resulted in undetectable viral titers in the lung after SARS-CoV-2 challenge. These pre-clinical results demonstrate that antigen-sparing greater than 10-fold can be achieved in a mouse model, providing a valuable input for clinical trials in humans. In a pilot hamster experiment, the inactivated NDV-S vaccine is also immunogenic, inducing high titers of spike-specific antibodies. Since hamsters are much more susceptible to SARS-CoV-2 infection, the group receiving the WT NDV lost up to 15% of its weight by day 5, while both NDV-S vaccinated groups with or without the adjuvant showed greatly attenuated weight loss and reduced viral titers in the lungs. The AddaVax adjuvant again enhanced vaccine-induced protection, resulting in weight loss at only 2 dpi of the group. We did not evaluate the adjuvant R-DOTAP, as the dosing was not well-determined for this model by the time of this study. However, R-DOTAP and additional adjuvants will be evaluated in combination with the inactivated NDV-S vaccine in future studies. Moreover, the presented studies were highly focused on humoral responses and protection against the SARS-CoV-2 challenge in mice and hamsters as a proof of principle for the inactivated NDV-based vaccine, in which T cells and cytokine responses were not analyzed. It will be important to examine CD8+ T cells, CD4+ T cells, and Th1/Th2 responses in future preclinical studies with and without the adjuvant of choice to evaluate the mechanisms of protection and safety.

We have shown promising protection by immunization with inactivated NDV-S in both mouse and hamster models. Even though sterilizing immunity might not always be induced, the trade-off for having an affordable and widely available effective vaccine that reduces the symptoms of COVID-19 should be much preferred over a high-cost vaccine that is limited to high-income populations. Most importantly, the egg-based production of an NDV-S vaccine requires only minor modifications to the current inactivated influenza virus vaccine manufacturing process. The cost of inactivated influenza virus vaccines (trivalent and quadrivalent) is in the low-dollar range. Since NDV grows to similarly high titers as the influenza virus, the cost of goods should be similar to that of a monovalent inactivated influenza virus vaccine (a fraction of the cost of a quadrivalent seasonal influenza virus vaccine), or even lower due to dose sparing with an adjuvant that is inexpensive to manufacture.

## 5. Conclusions

We conclude that the inactivated NDV-S vaccine administered through the intramuscular route is immunogenic, inducing potent antibody responses targeting the spike protein in mice and hamsters. The vaccine attenuated weight loss and reduced viral loads in the lungs of mice and hamsters after the SARS-CoV-2 challenge. In the presence of an adjuvant, antigen-sparing could be achieved, which would potentially further ensure the low-cost of the vaccine when produced using the existing influenza virus vaccine capacity.

## Figures and Tables

**Figure 1 vaccines-08-00771-f001:**
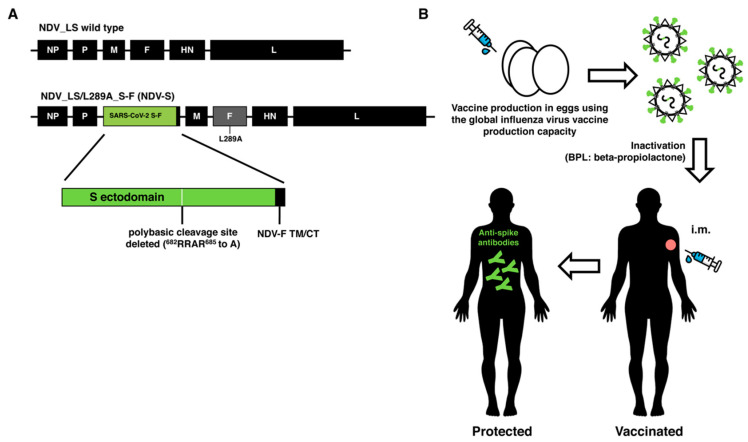
Design and concept of an inactivated Newcastle disease virus (NDV)-based severe acute respiratory syndrome coronavirus 2 (SARS-CoV-2) vaccine. (**A**) Design of the NDV-S vaccine. The sequence of the spike-fusion (S-F) chimera (green: ectodomain of S, and black: the transmembrane domain and cytoplasmic tail of the NDV F protein) was inserted between the P and M gene of the NDV LaSota (NDV_LS) strain L289A mutant (NDV_LS/L289A). NDV-S: NDV_LS/L289A_S-F. The polybasic cleavage site of the S was removed (^682^RRAR^685^ to A). (**B**) The concept overview of an inactivated NDV-based SARS-CoV-2 vaccine. The NDV-S vaccine could be produced using the current global influenza virus vaccine production capacity. Such an NDV-S vaccine displays abundant S proteins on the surface of the virions. The NDV-S vaccine could be inactivated by beta-propiolactone (BPL). The NDV-S vaccine could be administered intramuscularly (i.m.) to elicit protective antibody responses in humans.

**Figure 2 vaccines-08-00771-f002:**
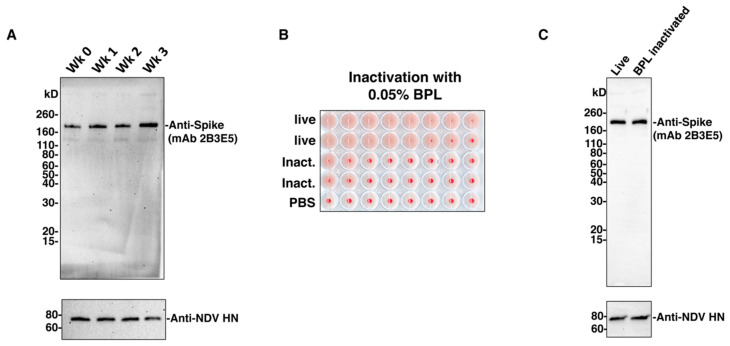
The S-F chimera is stable. (**A**) Stability of the S-F chimera at 4 °C. Allantoic fluid containing the NDV-S virus was aliquoted into equal amounts (15 mL) and stored at 4 °C. Virus from each aliquot was concentrated through a 20% sucrose cushion, re-suspended in an equal amount of phosphate buffered saline (PBS), and then stored at −80 °C for several weeks (wk 0, wk 1, wk 2, and wk 3). One microgram of each concentrated virus was resolved onto 4–20% sodium dodecyl sulfate polyacrylamide gel electrophoresis (SDS-PAGE). Protein degradation was evaluated by Western blot using the S-specific mouse monoclonal antibody 2B3E5. The hemagglutinin-neuraminidase (HN) protein of NDV was used as an NDV protein control. (**B**) Inactivation of the virus by beta-propiolactone (BPL). Viruses in the allantoic fluid were inactivated by 0.05% BPL, as described previously. Clarified allantoic fluids with live and inactivated viruses were diluted in PBS (at 1000-fold dilution) and inoculated into 10-day-old embryonated chicken eggs. The eggs were incubated at 37 °C for 3 days. The loss of infectivity of the inactivated virus was confirmed by the lack of growth of the virus determined by a hemagglutination (HA) assay. (**C**) Stability of the S-F before and after BPL inactivation. Live or inactivated (using 0.05% BPL) NDV-S virus was concentrated through a 20% sucrose cushion, as described previously. Two micrograms of live or BPL-inactivated virus were loaded onto 4–20% SDS-PAGE. Stability loss of the S-F was evaluated by Western blot, as described in A.

**Figure 3 vaccines-08-00771-f003:**
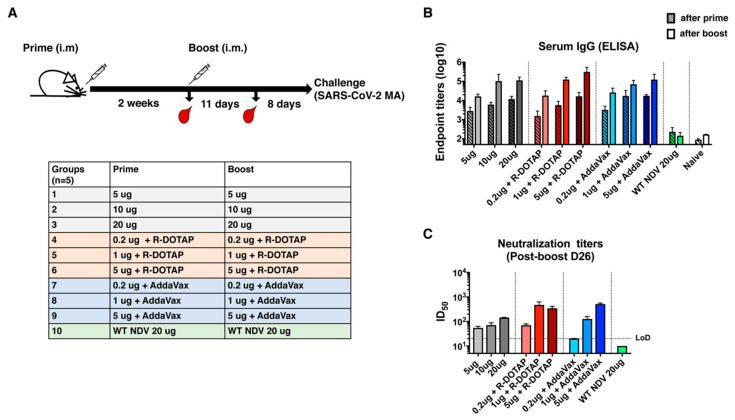
Inactivated NDV-S vaccine elicits high antibody responses in mice. (**A**) Immunization regimen and groups. BALB/c mice were given two immunizations via the intramuscular administration route with a 2-week interval. Mice were bled pre-boost and 11 days after the boost for in vitro serological assays. Mice were challenged with a mouse-adapted SARS-CoV-2 strain 19 days after the boost. Ten groups described in the table were included in this study. Group 1, 2, and 3 were immunized with 5, 10, and 20 μg of vaccine, respectively; group 4, 5, and 6 were immunized with 0.2, 1, and 5 μg of vaccine formulated with the R-enantiomer of the cationic lipid DOTAP (R-DOTAP), respectively; group 7, 8, and 9 were immunized with 0.2, 1, and 5 μg of vaccine combined with AddaVax, respectively; and group 10 was immunized with 20 μg of WT NDV virus as the vector-only control. (**B**) Spike-specific serum IgG titers. Serum IgG titers from animals after prime (pattern bars) and boost (solid bars) toward the recombinant trimeric spike protein were measured by an enzyme linked immunosorbent assay (ELISA). Endpoint titers were shown as the readout for ELISA. (**C**) Neutralization titers of serum antibodies. Microneutralization assays were performed to determine the neutralizing activities of serum antibodies from animals after the boost (D26) using the USA-WA1/2020 SARS-CoV-2 strain. The 50% of inhibitory dilution (ID_50_) of serum samples showing no neutralizing activity (WT NDV) was set as 10 (LoD: limit of detection).

**Figure 4 vaccines-08-00771-f004:**
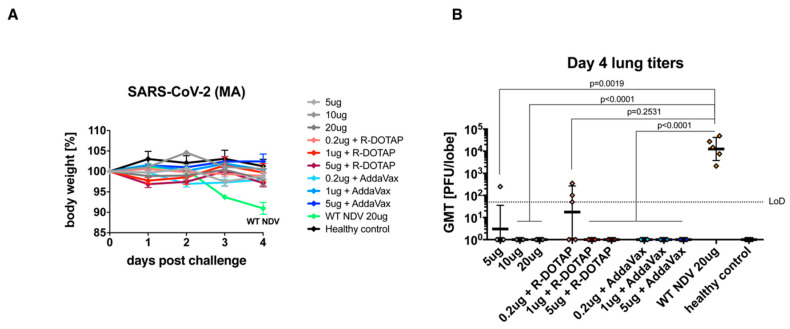
Inactivated NDV-S vaccine protects mice from SARS-CoV-2 infection. (**A**) Weight loss of mice infected with SARS-CoV-2. Weight loss of mice challenged with a mouse-adapted SARS-CoV-2 strain was monitored for 4 days. (**B**) Viral titers in the lung. Lungs of mice were harvested at day 4 post-infection. Viral titers of the lung homogenates were determined by a plaque assay. Geometric mean titer (PFU/lobe) is shown (LoD: limit of detection). Statistical analysis was performed using the Kruskal–Wallis test with Dunn’s correction for multiple comparisons. P-values between groups were shown.

**Figure 5 vaccines-08-00771-f005:**
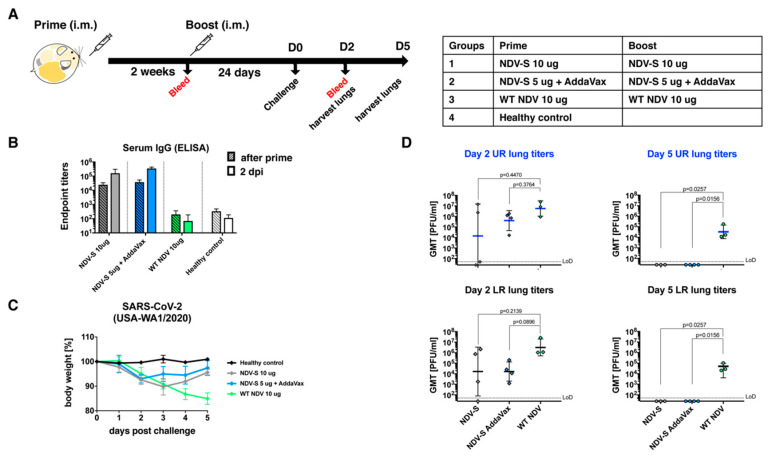
Inactivated NDV-S vaccine attenuates SARS-CoV-2 infection in hamsters. (**A**) Immunization regimen and groups. Golden Syrian hamsters were vaccinated with inactivated NDV-S following a prime-boost regimen with a 2-week interval. Hamsters were challenged 24 days after the boost with the USA-WA1/2020 SARS-CoV-2 strain. Four groups of hamsters (*n* = 8) were included in this study. Group 1 received 10 μg of inactivated NDV-S vaccine without any adjuvant. Group 2 received 5 μg of inactivated NDV-S vaccine adjuvanted with AddaVax. Group 3 receiving the 10 μg of inactivated WT NDV was included as the vector-only (negative) control. Group 4 animals receiving no vaccine were mock challenged with PBS as healthy controls. (**B**) Spike-specific serum IgG titers. Hamsters were bled pre-boost and a subset of hamsters were terminally bled at 2 days post-infection (dpi). Vaccine-induced serum IgG titers towards the trimeric spike protein were determined by ELISA. Endpoint titers are shown as the readout for ELISA. (**C**) Weight loss of hamsters challenged with SARS-CoV-2. Weight loss of SARS-CoV-2-infected hamsters was monitored for 5 days. (**D**) Viral titers in the lungs. Viral titers in the upper right (UR) and lower right (LR) lung lobes of the animals at 2 and 5 dpi were measured by a plaque assay (LoD: limit of detection). Statistical analysis was performed using the Kruskal–Wallis test with Dunn’s correction for multiple comparisons. P-values between groups were shown.

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
