# Peer review of "A Newcastle Disease Virus (NDV) Expressing a Membrane-Anchored Spike as a Cost-Effective Inactivated SARS-CoV-2 Vaccine"

_vaccines, 2020, doi:10.3390/vaccines8040771_

Round 1
Reviewer 1 Report
The authors have demonstrated proof of concept for SARS-CoV-2 vaccine by expressing S protein in NDV. The vaccine has shown reasonable protection in mice and Hamster model. There are few concerns that needs to be addressed for this manuscript for consideration:
- The neutralization results are encouraging however the authors did not provide any T-cell response data. Please include this aspect into the manuscript. Flow cytometric analysis will be helpful.
- Please provide cytokines response for the treatment groups.
- Include representative images (in supplementary materials) for the lung H&E staining.
- The impetus on “cost effective” vaccine is crucial however the authors have not presented any cost analysis for the vaccine and appears very generalized. Authors may consider appropriate title as NDV based vaccines is not new to the field and research community is aware of the significance. This decision is left to editor’s discretion.
For better evaluation and understanding of SARS-COV-2 vaccine efficacy, it is essential to have cytokine and T-Cell response, it is highly recommended to include this in the manuscript.
Author Response
Reviewer 1:
The authors have demonstrated proof of concept for SARS-CoV-2 vaccine by expressing S protein in NDV. The vaccine has shown reasonable protection in mice and Hamster model. There are few concerns that needs to be addressed for this manuscript for consideration:
- The neutralization results are encouraging however the authors did not provide any T-cell response data. Please include this aspect into the manuscript. Flow cytometric analysis will be helpful.
- Please provide cytokines response for the treatment groups.
Responses to comments 1 and 2: We thank the reviewer for the suggestions. We respectfully ask that work on T- cells responses to our vaccine would be part of a future paper. We also want to stress the fact that our NDV-S is a beta-propiolactone (BPL)-inactivated vaccine. In the past, such inactivated vaccines (eg. the inactivated influenza virus vaccine) have been shown to be low at inducing T-cell responses. In non-human primates, it was reported that neutralizing antibodies, but not T cells responses, correlated with protection (Yu, J. et al., Science, 2020). Therefore, because of low T-cell responses induced by vaccination, we did not include T-cell experiments in the present manuscript.
- Include representative images (in supplementary materials) for the lung H&E staining.
Response: We thank the reviewer for this comment. Due to the short period of time for revision and unavailability of samples, we are not able to address this reviewer’s request on additional histopathology data. In fact, because of the comments of the second reviewer, we had to remove all the histopathology data and the reference to it.
- The impetus on “cost effective” vaccine is crucial however the authors have not presented any cost analysis for the vaccine and appears very generalized. Authors may consider appropriate title as NDV based vaccines is not new to the field and research community is aware of the significance. This decision is left to editor’s discretion.
Response: To stress the “cost effective” aspect of the vaccine, we added more discussion (line 621-623)
- For better evaluation and understanding of SARS-COV-2 vaccine efficacy, it is essential to have cytokine and T-Cell response, it is highly recommended to include this in the manuscript.
Response: Please see “Responses to comments 1 and 2” above.
Reviewer 2 Report
The manuscript describes the development and preclinical testing of an inactivated NDV-S vector vaccine for SARS-CoV-2. Application of this vaccine with and without adjuvant was tested in laboratory mice and hamsters and showed that use of the adjuvant reduced the required viral dose substantially. The approach of the current study is promising as it offers perspectives for development of an inexpensive vaccine that can be manufactured under BSL-2 conditions instead of the BSL-3 conditions that are otherwise required for work with SARS-CoV-2. Another aspect that the authors indicate is that the global egg based influenza vaccine capacity could be used to produce this suggested vaccine, allowing large scale production at low cost.
The authors use a described NDV_LS/L289A_S-F rescue plasmid and part of the S gene of the Wuhan-Hu-1/2020 SARS-CoV-2 isolate as well as a mouse adapted SARS-CoV-2 strain for their study.
For the preclinical part of the study, the authors used 7-week-old female BALB/cJ mice and inoculated these with the mouse adapted SARS-CoV-2 strain. At 4 days post challenge they examined the viral loads. This was not accompanied by any histological examination. What is the reason for this? It would have been interesting to see the effect of the vaccine in particular also on the type, degree and composition of the inflammatory processes in response to the virus challenge (as they are, for example, reported in Refs. 2 and 21). The mouse lung is certainly large enough to allow all examinations including histology…
The authors used 8-week-old female golden Syrian hamsters as these have been shown to develop disease that resembles COVID-19. They have euthanised the animals at 2 and 5 days post challenge and collected lungs also for a histological examination. However, they only provide information on the histological features at 5 dpi. Unfortunately, the histological findings are not reported in a meaningful way (see also further comments below). This part of the study should be improved. It should preferably also include the results of the histological examination of the animals at 2 dpi as well as the immnohistological demonstration of SARS-CoV-2 antigen expression. These additional examinations would elucidate the potential effect of the vaccines on the pulmonary damage and the type, degree and composition of the inflammatory processes induced by SARS-CoV-2. The literature shows that the methods are well established in the labs of the author consortium, so these informative examinations could obviously be incorporated without much effort. Images to illustrate the findings would also be useful.
There are some specific issues that should be addressed. These are listed in a consecutive manner.
Line 209/210/211: Please reword the sentence to better reflect what was done, such as “Left lung lobes of the treated hamsters were collected and fixed in neutral-buffer formalin for at least 24 hours before being taken out of the BSL-3 facility and transferred to the Comparative Pathology Laboratory (CPL) at ISMMS where they were trimmed and routinely paraffin wax embedded. Sections (5 µm) were prepared and stained with hematoxylin and eosin (H&E).”
Also, the authors state here that they examined the lungs of “treated hamsters” (line 209) which implies that samples from all hamsters were examined. However, in the results sections they then state that they histologically examined the lungs of the hamsters euthanised at day 5. The results of the histological examination undertaken on the hamsters at 2 dpi should also be provided.
Results
Lines 252-269: What is now paragraph 3.1 with Figure 1 is not really appropriate as a paragraph in the Results section, since it does not provide any results. Instead, it would probably fit better as an introductory paragraph into the M&M section.
Line 370: Why did the authors determine the viral load/lobe? They apparently prepared a homogenate of all lung lobes together (see 2.9) which they homogenised. Why was this not adapted to a defined amount of tissue?
Line 370/371: A histological and immunohistological examination to detect SARS-CoV-2 antigen in the lungs of the mice had been useful as it would provide information on the changes and potential inflammatory response associated with the low viral titres in the animals immunised with 5 μg of unadjuvanted vaccine or 0.2 μg of vaccine with R-DOTAP.
Line 376-378: This final statement on the mouse experiment is maybe a bit too strong, as here a mouse adapted strain was tested in laboratory mice. It does of course indicate the potential of the inactivated NDV-S.
Line 435/436: The text “fixed in neutral-buffer formalin for at least 24 hours before being taken out of the BSL-3 facility” would be more appropriate in the M&M part (2.10); see previous comment.
Line 439: Reword “disease outcomes”, since, for example, weight loss is a sign of disease but not necessarily an outcome (which could be recovery or death).
Line 435: In the M&M section, the authors state that they examined the lungs of “treated hamsters” which implies that samples from all hamsters were examined. However, they now state that they histologically examined only the lungs of the hamsters euthanised at day 5. Why? At 2 dpi, they detected virus in the lungs. It would therefore be useful to add the results of the histological assessment as well (and complement these with immunohistology for SARS-CoV-2 antigen).
Line 439: The wording “substantial reduction of pathology” is meaningless without any description of the principal findings. The latter should be added.
Line 441: Should “least” not better be replaced by “lowest”?
Figure 6. In M&M the authors state that they scored the extent of epithelial degeneration and/or necrosis as well as the extent of each inflammation, hyperplasia and fibrosis. They should provide a reference for this scoring scheme, to show that it is appropriate. Also, they should specify whether they have assessed respiratory epithelial cell or alveolar epithelial cell necrosis or both.
At this stage (Fig. 6), they also introduce a “gross histopathological clinical score”. It is not clear which parameters comprise this score. From M&M (lines 212-216) one would assume that there are 4 components with a score of 0-4 each. In Figure 6 it is indicated that inflammation was scored twice, whereas epithelial degeneration/necrosis and fibrosis were not assessed at all. It is also not clear what is meant by a score for “type II pneumocyte hyperplasia/cytopathy” (NB: “cytopathy” is not a descriptive term in pathology). Furthermore, “perivascular inflammation” and “alveolar inflammation” are very vague terms that would need to be specified. Why did the authors split the “inflammation” into these two components? This would also need to be clarified.
Discussion
Line 463/363: Is the determination of weight loss and viral loads in the lungs sufficient to conclude that disease was attenuated?
Line 468: See comment on line 439. Please fill “the least extent of pathology” with some meaning, i.e. provide more information.
Author Response
The manuscript describes the development and preclinical testing of an inactivated NDV-S vector vaccine for SARS-CoV-2. Application of this vaccine with and without adjuvant was tested in laboratory mice and hamsters and showed that use of the adjuvant reduced the required viral dose substantially. The approach of the current study is promising as it offers perspectives for development of an inexpensive vaccine that can be manufactured under BSL-2 conditions instead of the BSL-3 conditions that are otherwise required for work with SARS-CoV-2. Another aspect that the authors indicate is that the global egg based influenza vaccine capacity could be used to produce this suggested vaccine, allowing large scale production at low cost.
The authors use a described NDV_LS/L289A_S-F rescue plasmid and part of the S gene of the Wuhan-Hu-1/2020 SARS-CoV-2 isolate as well as a mouse adapted SARS-CoV-2 strain for their study.
- For the preclinical part of the study, the authors used 7-week-old female BALB/cJ mice and inoculated these with the mouse adapted SARS-CoV-2 strain. At 4 days post challenge they examined the viral loads. This was not accompanied by any histological examination. What is the reason for this? It would have been interesting to see the effect of the vaccine in particular also on the type, degree and composition of the inflammatory processes in response to the virus challenge (as they are, for example, reported in Refs. 2 and 21). The mouse lung is certainly large enough to allow all examinations including histology…
Response: Many thanks to this reviewer who has excellent knowledge on the lung pathology of animals. In this manuscript, we mainly focus on antibody responses and protection of inactivated NDV-S in animal models against SARS-CoV-2 infection as a proof of principle. We would like to humbly point out that due to the limited time for revision and unavailability of samples, we are afraid we cannot address the reviewer’s comments on the requested histological data. Therefore, we have decided to remove all the histopathology data and the reference to it. We want to follow the reviewer’s advice to perform histological examinations thoroughly in future studies. We hope the reviewer would kindly re-evaluate the manuscript based on the data without histopathology.
- The authors used 8-week-old female golden Syrian hamsters as these have been shown to develop disease that resembles COVID-19. They have euthanised the animals at 2 and 5 days post challenge and collected lungs also for a histological examination. However, they only provide information on the histological features at 5 dpi. Unfortunately, the histological findings are not reported in a meaningful way (see also further comments below). This part of the study should be improved. It should preferably also include the results of the histological examination of the animals at 2 dpi as well as the immnohistological demonstration of SARS-CoV-2 antigen expression. These additional examinations would elucidate the potential effect of the vaccines on the pulmonary damage and the type, degree and composition of the inflammatory processes induced by SARS-CoV-2. The literature shows that the methods are well established in the labs of the author consortium, so these informative examinations could obviously be incorporated without much effort. Images to illustrate the findings would also be useful.
Response: As indicated under 1, we decided to focus on antibody responses and protection provided by our inactivated NDV-S vaccine. We eliminated all the data on histopathology and the reference to it.
There are some specific issues that should be addressed. These are listed in a consecutive manner.
- Line 209/210/211: Please reword the sentence to better reflect what was done, such as “Left lung lobes of the treated hamsters were collected and fixed in neutral-buffer formalin for at least 24 hours before being taken out of the BSL-3 facility and transferred to the Comparative Pathology Laboratory (CPL) at ISMMS where they were trimmed and routinely paraffin wax embedded. Sections (5 µm) were prepared and stained with hematoxylin and eosin (H&E).”
Response: We have removed all the data on histopathology and the reference to it.
- Also, the authors state here that they examined the lungs of “treated hamsters” (line 209) which implies that samples from all hamsters were examined. However, in the results sections they then state that they histologically examined the lungs of the hamsters euthanised at day 5. The results of the histological examination undertaken on the hamsters at 2 dpi should also be provided.
Response: We have removed all the data on histopathology and the reference to it.
- Results
Lines 252-269: What is now paragraph 3.1 with Figure 1 is not really appropriate as a paragraph in the Results section, since it does not provide any results. Instead, it would probably fit better as an introductory paragraph into the M&M section.
Response: We thank the reviewer for the comment. We have decided to move and adapt the original 3.1 and figure 1 into the introduction (line 57-111).
- Line 370: Why did the authors determine the viral load/lobe? They apparently prepared a homogenate of all lung lobes together (see 2.9) which they homogenised. Why was this not adapted to a defined amount of tissue?
Response: We thank the reviewer to point this out. Only the inferior lobes (lowest lobe on the right side) of mice were collected as a routine procedure for the mouse model used. The lobe was homogenized in 1 ml PBS. We have clarified this information in “Materials and Methods” under “2.10 lung titers” (line 286)
- Line 370/371: A histological and immunohistological examination to detect SARS-CoV-2 antigen in the lungs of the mice had been useful as it would provide information on the changes and potential inflammatory response associated with the low viral titres in the animals immunised with 5 μg of unadjuvanted vaccine or 0.2 μg of vaccine with R-DOTAP.
Response: As indicated under 1, we have removed all the data on histopathology and the reference to it.
- Line 376-378: This final statement on the mouse experiment is maybe a bit too strong, as here a mouse adapted strain was tested in laboratory mice. It does of course indicate the potential of the inactivated NDV-S.
Response: We removed the statement “To conclude, the inactivated NDV-S exhibits great potential as a cost-effective vaccine inducing protective immunity against the SARS-CoV-2 at very low doses with an adjuvant.” in 3.3 section at line 506.
- Line 435/436: The text “fixed in neutral-buffer formalin for at least 24 hours before being taken out of the BSL-3 facility” would be more appropriate in the M&M part (2.10); see previous comment.
Response: We have removed all the data on histopathology and the reference to it.
10 Line 439: Reword “disease outcomes”, since, for example, weight loss is a sign of disease but not necessarily an outcome (which could be recovery or death).
Response: We will not use the term “disease outcomes”. It was removed together with all the histological data.
- Line 435: In the M&M section, the authors state that they examined the lungs of “treated hamsters” which implies that samples from all hamsters were examined. However, they now state that they histologically examined only the lungs of the hamsters euthanised at day 5. Why? At 2 dpi, they detected virus in the lungs. It would therefore be useful to add the results of the histological assessment as well (and complement these with immunohistology for SARS-CoV-2 antigen).
Response: We have removed all the data on histopathology and the reference to it
- Line 439: The wording “substantial reduction of pathology” is meaningless without any description of the principal findings. The latter should be added.
Response: The sentence is removed together with histopathology data.
- Line 441: Should “least” not better be replaced by “lowest”?
Response: The word is removed together with histopathology data.
- Figure 6. In M&M the authors state that they scored the extent of epithelial degeneration and/or necrosis as well as the extent of each inflammation, hyperplasia and fibrosis. They should provide a reference for this scoring scheme, to show that it is appropriate. Also, they should specify whether they have assessed respiratory epithelial cell or alveolar epithelial cell necrosis or both.
Response: We have removed the sub-section on histopathology in “Materials and Methods”
- At this stage (Fig. 6), they also introduce a “gross histopathological clinical score”. It is not clear which parameters comprise this score. From M&M (lines 212-216) one would assume that there are 4 components with a score of 0-4 each. In Figure 6 it is indicated that inflammation was scored twice, whereas epithelial degeneration/necrosis and fibrosis were not assessed at all. It is also not clear what is meant by a score for “type II pneumocyte hyperplasia/cytopathy” (NB: “cytopathy” is not a descriptive term in pathology). Furthermore, “perivascular inflammation” and “alveolar inflammation” are very vague terms that would need to be specified. Why did the authors split the “inflammation” into these two components? This would also need to be clarified.
Response: We have removed Fig. 6 and Materials and Methods section on histopathology.
Discussion
- Line 463/363: Is the determination of weight loss and viral loads in the lungs sufficient to conclude that disease was attenuated?
Response: We thank the reviewer for the comment. To ensure accuracy, we have changed/removed the term “disease was attenuated” and its equivalent expressions (line 29, 121, 554, 583).
- Line 468: See comment on line 439. Please fill “the least extent of pathology” with some meaning, i.e. provide more information.
Response: The sentence is removed together with all the histopathology data.
Round 2
Reviewer 1 Report
The authors’ response (to previous comment regarding) inclusion of T-cell response is self-contradicting and confusing. It is unclear how the authors are going to have T cell response data for a future manuscript with their vaccine candidate if they expect low T cell response and also provide some reference for it. It would be prudent to include it in this manuscript in either case.
It is not always true that beta-propiolactone inactivated vaccine induce low T cell response it depends on the vaccine design; the vaccine candidate screening and selection is in such a way that they induce a desired balanced response. Also, the inactivation process for Influenza virus, NDV or SARS-COV-2 will be different and so is the immune response. That is why it is crucial to evaluate the T-cell response (tilt for CD4 and CD8) SARS-COV-2 is notorious for more damage due cytokine storm so a T cell and cytokine response is critical. T cell response is durable and gives faster and strong anti-viral response. Refer JAMA. 2020;324(13):1279-1281. doi:10.1001/jama.2020.16656; Nature volume 584, pages457–462(2020); Gao et al., Science 369, 77–81 (2020); Nature Reviews Immunology volume 20, pages709–713(2020); Science Immunology DOI: 10.1126/sciimmunol.abd6160.
The authors refer to a DNA vaccine candidate(s) to make a point that neutralizing Abs offers protection and not T-cell response (Yu, J. et al., Science, 2020), however this example is not valid to this study as it is not inactivated vaccine. Also, such characteristic of inactivated vaccine candidate (without T cell and cytokine response) is not appreciated for further pre- or clinical studies. Of course the neutralization is important but T-cell response is equally important.
The authors have used 2 types of adjuvants at varying dose of vaccine, it is not clear which neutralizing vaccine dose and adjuvant is safe, mere H&E score is insufficient to conclude. However now in the revised manuscript even the histology data is removed, it is not clear why it was removed? Did the other Reviewer or this reviewer recommend it! Is neutralizing Ab titer the only matrix for screening the vaccine candidate? The present study is incomplete.
The authors have not addressed any of the critical suggestions/comments raised to improve the manuscript.
Authors have reasoned “short period of time for revision”: I am sure the Journal editors are considerate enough to grant the authors reasonable time for revision, did authors communicate with the Journal regarding this? This reviewer will bring this issue to the Editor and the Journal office.
Authors have reasoned “unavailability of samples”: It is then recommended to perform the experiment again.
Overall, the study is incomplete and lacks critical data to infer and is not recommended for publication in the present form.
Author Response
Reviewer 1
The authors’ response (to previous comment regarding) inclusion of T-cell response is self-contradicting and confusing. It is unclear how the authors are going to have T cell response data for a future manuscript with their vaccine candidate if they expect low T cell response and also provide some reference for it. It would be prudent to include it in this manuscript in either case.
It is not always true that beta-propiolactone inactivated vaccine induce low T cell response it depends on the vaccine design; the vaccine candidate screening and selection is in such a way that they induce a desired balanced response. Also, the inactivation process for Influenza virus, NDV or SARS-COV-2 will be different and so is the immune response. That is why it is crucial to evaluate the T-cell response (tilt for CD4 and CD8) SARS-COV-2 is notorious for more damage due cytokine storm so a T cell and cytokine response is critical. T cell response is durable and gives faster and strong anti-viral response. Refer JAMA. 2020;324(13):1279-1281. doi:10.1001/jama.2020.16656; Nature volume 584, pages457–462(2020); Gao et al., Science 369, 77–81 (2020); Nature Reviews Immunology volume 20, pages709–713(2020); Science Immunology DOI: 10.1126/sciimmunol.abd6160.
The authors refer to a DNA vaccine candidate(s) to make a point that neutralizing Abs offers protection and not T-cell response (Yu, J. et al., Science, 2020), however this example is not valid to this study as it is not inactivated vaccine. Also, such characteristic of inactivated vaccine candidate (without T cell and cytokine response) is not appreciated for further pre- or clinical studies. Of course the neutralization is important but T-cell response is equally important.
The authors have used 2 types of adjuvants at varying dose of vaccine, it is not clear which neutralizing vaccine dose and adjuvant is safe, mere H&E score is insufficient to conclude. However now in the revised manuscript even the histology data is removed, it is not clear why it was removed? Did the other Reviewer or this reviewer recommend it! Is neutralizing Ab titer the only matrix for screening the vaccine candidate? The present study is incomplete.
The authors have not addressed any of the critical suggestions/comments raised to improve the manuscript.
Authors have reasoned “short period of time for revision”: I am sure the Journal editors are considerate enough to grant the authors reasonable time for revision, did authors communicate with the Journal regarding this? This reviewer will bring this issue to the Editor and the Journal office.
Authors have reasoned “unavailability of samples”: It is then recommended to perform the experiment again.
Overall, the study is incomplete and lacks critical data to infer and is not recommended for publication in the present form.
Response: We fully understand Reviewer 1 and agree with him/her that the study is incomplete without a careful T- cell analysis. However, we respectfully ask that this paper is only judged on the data showing the humoral responses and protection conferred by our vaccine approach. Most importantly, we present not only the protection study in a mouse model using the mouse-adapted SARS-CoV-2 strain (the gold standard for challenge studies, right now), but also – in parallel- a protection study in hamsters using the wild type SARS-CoV-2. We really feel we could not do justice to a T-cell analysis unless we do a thorough investigation going beyond the scope of the present manuscript.
Again, we ask this paper is only judged on the merits of these very focused sets of data. After all, we are just starting to develop an inactivated NDV-based vector vaccine against COVID-19. To our knowledge, no other group has studied this approach. We believe that it is a safe, effective and low-cost strategy for a second generation of COVID-19 vaccines
We have removed any mention of T-cells immunity and the possible mechanism by which our inactivated NDV-S vaccine works, agreeing with the reviewer that such a comprehensive analysis will be important in the future to fully understand this strategy (lines 336-337).
To address the last point of the reviewer, we stress in the text that the choice of the vaccine dose and adjuvant is only based on the in vitro neutralizing results “A dose-ranging study of the vaccine in the presence or the absence of an adjuvant was based on the ability to induce antibody/neutralizing antibody responses.” (lines 309-311).
Again, we ask that this paper is not judged as the ultimate all-encompassing study on the NDV vector; this is the beginning of our ongoing development of an NDV-based COVID-19 vaccine.
We believe it’s not fair to request such an exhaustive treatment of the topic for this provisionary manuscript.
Reviewer 2 Report
The manuscript has now been “cleaned up” in so far as the authors have removed the entire histology part. While this is a pity in my opinion, I accept the decision due to the shortage of time for revision. All other matters were adequately addressed. Thanks you.
There are only a few minor issues that the authors should address:
Line 94/95: Can one state that the “animals were protected” when they only showed less weight loss and lower viral loads? It would be better to reword this and say that they were protected to some extent, but obviously not fully, since several still became infected.
Lines 435-436: Is there a word missing in this part of the sentence?
Line 455: Same comment as re. line 94/95.
Author Response
Reviewer 2
The manuscript has now been “cleaned up” in so far as the authors have removed the entire histology part. While this is a pity in my opinion, I accept the decision due to the shortage of time for revision. All other matters were adequately addressed. Thanks you.
Response: We very much thank the reviewer for his/her understanding
There are only a few minor issues that the authors should address:
Line 94/95: Can one state that the “animals were protected” when they only showed less weight loss and lower viral loads? It would be better to reword this and say that they were protected to some extent, but obviously not fully, since several still became infected.
Response: We have removed the conclusion “animals were protected” and rephrased the sentence to “The vaccinated animals showed less weight loss and significantly reduced viral loads in the lungs as compared to the vector-only control animals after being challenged with a mouse-adapted SARS-CoV-2 strain” (line 95 and 96)
Lines 435-436: Is there a word missing in this part of the sentence?
Response: We thank the reviewer to point this out. We added “showed” to the sentence “…while both NDV-S vaccinated groups with or without the adjuvant showed greatly attenuated weight loss and reduced viral titers in the lungs” (line 444-445)
Line 455: Same comment as re. line 94/95.
Response: We have rephrased the sentence to “The vaccine attenuated weight loss and reduced viral loads in the lungs of mice and hamsters after the SARS-CoV-2 challenge” (line 464-467)
Round 3
Reviewer 1 Report
The authors have failed to address any comments and concerns brought to their attention. The authors agree with the reviewer on the concerns (for both revisions) and it is surprising that there were no attempts to address it and instead have given unsatisfactory excuses and some of their responses have been self-contradictory.
The authors have already published (also cited in the manuscript) a similar research with NDV-SARS-CoV-2 (live) in EBioMedicine there is no exploration of cytokine, T cell response and safety aspect. Therefore, it is quite natural to expect these aspects in this manuscript.
It is not clear why the authors are reluctant to perform any experiments to justify the work. Actually, for few concerns, there is no need to perform the whole experiment again as the lung homogenates and bleeds of previous experiments can be used for performing ELISA for important cytokines (e.g. Luminex/bioplex).
The authors have not made any attempts to include limitations of the study, do they not think it is necessary to mention as there are many concerns with the study.
Author Response
Please see cover letter to the editor-in-chief